# Maternal Exposure to Cigarette Smoke during Pregnancy and Testicular Cancer in Offspring: A Systematic Review and Meta-Analysis

**DOI:** 10.3390/life13030618

**Published:** 2023-02-23

**Authors:** Astrid L. Beck, Elvira V. Bräuner, Russ Hauser, Youn-Hee Lim, Cecilie S. Uldbjerg, Anders Juul

**Affiliations:** 1Department of Growth and Reproduction, Copenhagen University Hospital—Rigshospitalet, 2100 Copenhagen, Denmark; 2International Centre for Research and Research Training in Endocrine Disruption of Male Reproduction and Child Health (EDMaRC), Copenhagen University Hospital—Rigshospitalet, 2100 Copenhagen, Denmark; 3Harvard T.H. Chan School of Public Health, Harvard University, Cambridge, MA 02138, USA; 4Section of Environmental Health, Department of Public Health, University of Copenhagen, 1014 Copenhagen, Denmark; 5Department of Preventive Medicine, Seoul National University College of Medicine, Seoul 03080, Republic of Korea; 6Department of Clinical Medicine, University of Copenhagen, 2200 Copenhagen, Denmark

**Keywords:** cigarette smoke, prenatal exposure, testicular cancer, systematic review, meta-analysis, epidemiology

## Abstract

Background: Maternal exposure to cigarette smoke in pregnancy may play a role in the development of testicular cancer in offspring. An updated and comprehensive systematic review of the available evidence is needed. Objective: To identify and evaluate current evidence on maternal exposure to cigarette smoke during pregnancy and testicular cancer in offspring. Methods: A systematic search of English peer-reviewed original literature in PubMed through a block search approach. Publications were considered if assessing maternal exposure to cigarette smoke and the risk of testicular cancer in offspring. Results: Among the 636 identified records, 14 publications were eligible for review and 10 for meta-analysis. Quality assessment of the publications was conducted. Most included publications were case-control studies (*n* = 11, 79%), while the remaining were ecological studies (*n* = 3, 21%). Completeness of reporting was high, but more than half were considered subject to potential bias. The trend synthesis showed that half (*n* = 7) of the included publications demonstrated a higher risk of testicular cancer in the sons of mothers exposed to cigarette smoke during pregnancy. The meta-analysis generated an overall summary risk estimate of 1.00 (95% CI: 0.88; 1.15) (*n* = 10 publications), with a lower risk for seminoma (0.79, 95% CI: 0.59; 1.04) and nonseminoma (0.96, 95% CI: 0.74; 1.26) (*n* = 4 publications). Conclusions: This systematic review did not provide evidence of an association between maternal exposure to cigarette smoke and risk of testicular cancer in offspring. An overall positive trend was suggested, but it had low statistical precision. The methodological limitations across publications encourage further research based on valid exposure data.

## 1. Introduction

Maternal exposure to cigarette smoke during pregnancy proposes a significant health risk to the mother and fetus and is a plausible risk factor for cancers [1]. The chemical constituents of cigarette smoke can pass through the placental barrier to the fetus, through which the developing fetus is subsequently exposed to central components of cigarette smoke, nicotine, and carbon monoxide; compounds known to cause contractions of placental vessels and fetal hypoxemia [1,2,3]. Despite a general reduction in the prevalence of cigarette smoking worldwide due to increasing awareness of the associated health risks [4], cigarette smoking among pregnant women remains surprisingly prevalent [5] and their exposure to secondhand smoke even more so [5].

As pioneered by Clemmesen almost three decades ago, prenatal exposure to cigarette smoke is a potential health risk proposed to be related to testicular cancer in offspring [6]. Testicular cancer is the growth of cancerous cells in the testis and predominantly affects young males aged 15 to 44 years [7,8]. The incidence of testicular cancer has increased worldwide during the past decades, with the causes unknown [9]. Genetic factors do not fully explain this trend and it has been suggested that lifestyle and the environment in modern society may play an important role [10]. This concept has been supported by geographical differences in the incidences of testicular cancer, with some of the highest incidence rates observed in the Nordic countries (Figure 1) [9,11,12,13], and moreover disproportionately affecting high-income countries more than that of low-income countries [9,14]. Furthermore, differential risks of the malignancy have been observed among first- and second-generation immigrants, reflecting the environment’s influence [15,16]. The young age of testicular cancer patients further aid speculations that environmental insults can act as early as *in utero* [10,17].

Several epidemiological studies have already linked maternal cigarette smoking with outcomes associated with testicular cancer, such as cryptorchidism and hypospadias [9,10,18,19,20], but the available evidence specifically on testicular cancer is rather sparse. The latest systematic review based on a literature search from 2013 concluded that there was no evidence of an association between maternal cigarette smoking and testicular cancer. However, the review lacked a comprehensive discussion of the methodological limitations of the included studies and a critical assessment of risk of bias.

We conducted a systematic review on maternal exposure to cigarette smoke during pregnancy and risk of testicular cancer in offspring to gain an overview of the existing scientific evidence to date. Moreover, as methodological propositions provide a foundation for substantive inference, we conducted individual evaluations of risk of bias to provide future recommendations for research.

## 2. Methods

### 2.1. Study Design

In accordance with the PRISMA guidelines for systematic reviews [21], a systematic literature review was conducted. In addition, a meta-analysis was performed to provide an overall weighted measure of association.

### 2.2. Protocol and Registration

The review protocol was registered at PROSPERO (registration number: CRD42022354600) and approved by all authors preceding the systematic literature search.

### 2.3. Search Strategy

To identify available epidemiological evidence assessing maternal exposure to cigarette smoke during pregnancy and its association with testicular cancer in offspring, we performed a systematic search of English peer-reviewed original literature on 14 September 2022. The literature search was conducted in the PubMed database through a block search approach where both MeSH and free-text search terms were utilized for search words. The identified search terms were divided into three blocks: (1) exposure (maternal exposure to cigarette smoke), (2) outcome (testicular cancer in offspring), and (3) window of exposure (prenatal period). All conceivable terms related to cigarette smoke exposure were included in Block 1 to fully encapsulate modern alternatives to cigarettes (e.g., “e-cigarettes” and “vaping”); however, the focus remained on cigarette smoke as this combined exposure to nicotine, tobacco, and other compounds in cigarettes. An auxiliary search was conducted in the EMBASE database due to its biomedical relevancy, hereunder following the same search structure as the primary search in PubMed, using Emtree terms and keywords. Lastly, a snowball search of the reference lists of included publications was conducted to include as many relevant publications as possible. The search specifications and the number of respective hits in each of the three blocks are provided in the search protocol available in Appendix A.

### 2.4. Eligibility Criteria

Maternal exposure to cigarette smoke during pregnancy. Maternal exposure to cigarette smoke was defined as prenatal exposure to cigarette smoke either by the mother’s own smoking or the mother’s exposure to passive cigarette smoke (e.g., in household or workplace). This exposure could either be ascertained through self-administered questionnaires, biological sampling, or interviewing.

Testicular cancer in offspring. Outcome was classified as testicular cancer and/or histological subtypes of testicular cancer (seminoma and nonseminoma) ascertained by either medical standardized examination, medical records, pathology slides, reporting to health registries, registration at cancer treatment centers, or self-reporting.

### 2.5. Exclusion Criteria

Publications covering non-human studies, case-reports, experimental studies, narrative reviews, systematic reviews, opinion pieces (e.g., editorials and comments), and non-relevant exposures or outcomes were excluded. Moreover, publications not in English or not providing a measure of an association were also excluded. The reasons for exclusion at the full text review screening stage can be viewed in the PRISMA flow diagram (Figure 1).

### 2.6. Selection of Literature

The systematic literature search resulted in 636 records from the combined PubMed and Embase database search. We obtained 386 records after the removal of duplicates. Two authors (A.L.B and C.S.U.) screened the titles and abstracts independently to assess relevancy to the research aim. Among these, 314 were beyond the context of the eligibility criteria, resulting in 72 records being eligible for full-text screening. Through the full-text screening, 58 records were excluded as they did not fulfill the eligibility criteria. Hand searches of the bibliographies of the relevant retrieved publications did not capture additional literature. Attempts were made to obtain unpublished data but did not result in additional literature. A total of 14 publications were eventually included for the quality assessment, the qualitative trend synthesis, and meta-analysis (Figure 2).

### 2.7. Quality Assessment

All 14 publications were evaluated according to their completeness of reporting and risk of bias by referring to a standardized form adapted from Bonzini et al. (2007) and Shamliyan et al. (2010) (Appendix A) [22,23]. Evaluation was conducted independently by two authors (A.L.B and C.S.U.) with any discrepancies resolved by a third author (E.V.B.). Preceding the evaluation, all three authors tested the form prior to the quality assessment to ensure exact mutual understanding of the terms and concepts to warrant reliability.

Completeness of reporting was assessed in the following nine areas: (1) study design, (2) sampling frame and procedures, (3) inclusion and exclusion, (4) population characteristics of exposed/unexposed or cases/referents, (5) response rate reported or implicitly given, (6) methods for exposure measurement, (7) method for outcome ascertainment, (8) statistical analysis, and (9) exposure–response. The nine areas were equally weighted with a value of 1 given for adequate reporting and a value 0 for inadequate reporting. We considered a sum score of ≥6 points as a sufficient completeness of reporting.

Potential sources of bias were evaluated in a total of eight areas, of which the following five areas were considered the most critical sources of bias and were included for final assessment: (i) reporting of tested hypotheses, (ii) selection bias from loss of follow-up or lack of representativeness in a population sample, (iii) information bias related to exposure ascertainment, (iv) information bias related to outcome ascertainment, and (v) accounting for confounding. Each of the five areas were either rated as high risk (score = 2), uncertain risk (score = 1), or low risk (score = 0). Individual publications were considered at high risk if the sum risk of bias score ≥ 8.

The assigned quality assessment scores for completeness of reporting and risk of bias for individual publications are presented in Appendix A.

### 2.8. Data Extraction and Qualitative Trend Synthesis

Descriptive characteristics were extracted from each included publication, hereunder author and publication year; study design, period, and population; country of population origin; ascertainment of both exposure and outcome; and whether the study assessed the histological subtype of testicular cancer.

To provide an overview of the direction of the association between maternal exposure to cigarette smoke during pregnancy and testicular cancer in offspring, a qualitative trend synthesis was conducted, illustrated by arrows: a downwards arrow indicating lower risk of testicular cancer in offspring (estimate below 1.00); upward arrow indicating higher risk of testicular cancer in offspring (estimate above 1.00); and a horizontal arrow indicating no association found (estimate equal to 1.00). An asterisk was added if the estimates were statistically significant.

### 2.9. Meta-Analysis

Publications were deemed eligible for the meta-analysis if they provided a comparable risk estimate (e.g., risk ratio and odds ratio) and 95% confidence interval (CI) between exposure and outcome. In the case of the sole report of several levels of exposure with no overall estimate provided, the highest level versus the reference category was chosen for further analysis. In the case of publications presenting both crude and confounder adjusted estimates, the latter was selected.

A forest plot for the meta-analysis was performed in Review Manager (RevMan v. 5.4, The Cochrane Collaboration, 2020) by utilizing an inverse-variance random-effects model to calculate an overall risk estimate. Heterogeneity (*I*^2^) statistics were calculated to evaluate in-between study variation and 25% was considered to be the threshold for importance of chance. A funnel plot by the standard error to the logarithm of the risk estimate was furthermore generated to assess sources of publication bias using the Egger’s regression-based test [24].

## 3. Results

### 3.1. Study Base

Selected characteristics of the 14 included publications reporting on the association between maternal exposure to cigarette smoke and risk of testicular cancer (and histological sub-groups, *n* = 8 publications) in offspring are presented in Table 1 [25,26,27,28,29,30,31,32,33,34,35,36,37,38]. The included publications were published in the period from 1979 to 2009 and were all based on Western populations. Most publications utilized a case-control study design (*n* = 11), while the remaining were ecological studies (*n* = 3).

Ascertainment of maternal exposure to cigarette smoke varied across the publications. Of the 11 case-control studies, one study measured cotinine concentrations in maternal serum collected during pregnancy, while the remaining 10 case-control studies based their ascertainment on self-report, hereunder self-administered questionnaires (*n* = 6) and interviews (*n* = 4), which were either ascertained retrospectively post-birth (*n* = 8) or during the first half of pregnancy (*n* = 2). Most case-control studies utilized binary exposures of ever/never having smoked during pregnancy (*n* = 7), while one study utilized a binary exposure with the cut-off point of seven cigarettes (≥7, <7 cigarettes smoked per day during pregnancy). The last two case-control studies utilized exposure levels with number of cigarettes smoked during pregnancy (0, 1–11, 12+ cigarettes) (*n* = 1) or a combination of binary exposures and exposure levels (*n* = 1). Of the three included ecological studies, two used tobacco-related lung cancers among the maternal generation as a proxy for smoke exposure, while the remaining ecological study utilized population level smoking behavior data of the presumed maternal generation (*n* = 1). The ecological studies ascertained smoking exposure through cancer registries (*n* = 2) and survey data (*n* = 1).

Diagnoses of testicular cancer in offspring were mainly extracted from national/regional cancer registries (*n* = 9) and through patient enrollment at cancer centers (*n* = 2) or cancer surveillance programs (*n* = 2). A combination of extracting information from both cancer registries and cancer centers was used in a single publication (*n* = 1).

Of the 14 identified publications, most had a sufficient completeness of reporting score (*n* = 12), and more than half of the publications were considered to have a high risk of bias (*n* = 8), according to our predefined criteria (Figure 3).

### 3.2. Qualitative Trend Synthesis (n = 14 Publications)

The results from the qualitative trend synthesis are presented in Table 1. Of the 14 publications, half (*n* = 7) demonstrated a trend toward a higher risk of testicular cancer in offspring whose mothers were exposed to cigarette smoke during their pregnancy compared with mothers who were not or who were less exposed. The remaining publications reported either a reduced risk (*n* = 3) or null estimates (*n* = 4) for associations with cigarette smoke during pregnancy.

Case-control studies (*n* = 11). A mix of both positive, negative, and no associations were observed across the included case-control studies, of which none reported statistical significance. Tuomisto et al. (2009) demonstrated that women with cotinine concentrations in the serum equivalent to that of a smoker (defined as ≥15 ng/mL) had a 32% lower risk of having a son with testicular cancer in comparison with women with non-smoker levels of cotinine (defined as <5 ng/mL) [37]. The case-control studies assessing self-reported data of exposure provided inconsistent findings. Coupland et al. (2004) [26] and Brown et al. (1986) [25] reported a 16% and 30% higher risk of testicular cancer in offspring if the mother reported smoking during their pregnancy, while Pettersson et al. (2007) [34] and Møller et al. (1996) [32] observed a 9% and 3% lower risk, respectively. Mongraw-Chaffin et al. (2009) [31] observed that mothers who smoked seven or more cigarettes a day during their pregnancy had a 6% higher risk of their son being diagnosed with testicular cancer, compared with women who smoked less than seven cigarettes a day. In the single study utilizing three exposure levels, Weir et al. (2000) [38] found no dose response, and sons born to mothers who smoked 1–11 cigarettes per day during pregnancy had a 10% higher risk of testicular cancer, while sons born to mothers who smoked ≥ 12 cigarettes per day had a 40% lower risk compared with sons born to mothers who did not smoke at all. The remaining four case-control studies by Henderson et al. (1987) [28], McGlynn et al. (2006) [30], Sonke et al. (2007) [35], and Swerdlow et al. (1987) [36] reported estimates at unity.

Ecological studies (*n* = 3). All three ecological studies showed a trend toward a higher risk of testicular cancer according to proxies of maternal exposure to cigarette smoking in pregnancy. Hemminki et al. (2005) [27] observed a 32% non-statistically significant higher risk of testicular cancer in sons born to mothers diagnosed with lung cancer, while Kaijser et al. (2003) [29] reported a 90% statistically significant higher risk. Pettersson et al. (2004) [34] observed geographical and temporal correlations between population level smoking habits in a cohort of women born between 1910 and 1940 and population level testicular cancer in males born between 1938 and 1968 years for Sweden (r = 0.99), Norway (r = 0.85), Denmark (r = 0.86), and Finland (r = 0.31), all of which were statistically significant except for Finland.

### 3.3. Meta-Analysis (n = 10 Publications)

Ten publications were considered eligible for the meta-analysis [25,26,30,31,32,33,35,36,37,38] (Table 1). Combined, these publications provide an overall summary risk estimate of 1.00 (95% CI, 0.88; 1.15) (Figure 4). The meta-associations of publications assessing histological subtypes (*n* = 4) generated estimates below unity for both seminoma (RR 0.79, 95% CI, 0.59; 1.04) and nonseminoma testicular cancer (RR 0.96, 95% CI, 0.74; 1.26) (Appendix A).

A funnel plot of all publications included in the meta-analyses assessing maternal exposure to cigarette smoke and testicular cancer in offspring indicated no publication bias (Figure 5), confirmed by the Egger’s regression-based test (−0.207, P_Begg’s Rank Correlation_ = 0.357).

## 4. Discussion

### 4.1. Overall Findings

In this systematic review, we conducted a meta-analysis of available human epidemiological evidence on the association between maternal exposure to cigarette smoke and testicular cancer in offspring, while assessing study quality and providing summary estimates of associations. On balance, a positive association was consistently observed in large ecological studies, while smaller case-control studies reported inconsistent associations and were hampered by a lack of statistical precision. Only one small case-control study quantified cotinine in maternal serum during pregnancy. Completeness of reporting of outcomes was high, but over 50% of the included publications were at potential risk of bias, specifically information and confounding bias.

### 4.2. Overall Findings from Qualitative Trend Synthesis and Meta-Analysis

The qualitative trend synthesis provided some indication of a trend toward a higher risk of testicular cancer if mothers were exposed to cigarette smoking during their pregnancy, but the overall pattern was largely inconsistent and only half of the publications demonstrated a risk estimate above 1.00, with most estimates being relatively close to unity and not statistically significant. Considering only the case-control studies, the results were particularly inconsistent with a mix of both positive, negative, and null associations, challenging the overall interpretation of findings. In contrast, all ecological cohort studies reported positive associations, with two detecting statistical significance. However, the ecological studies reporting statistical significance utilized maternal lung cancer and the smoking habits of the assumed maternal generation as a proxy for maternal cigarette smoking, thus cautious interpretation of these results is encouraged. On balance, the inconsistency in findings and general lack of statistical significance across the included publications subsequently impeded this current review from providing substantial conclusion.

The meta-analysis suggests that there is no association between maternal exposure to cigarette smoking and risk of testicular cancer in offspring; however, the observed risk estimate of 1.00 may also reflect the beforementioned inconsistent findings, thus nullifying the meta-analysis estimate. Unexpectedly, the association between maternal exposure to cigarette smoke was associated with a lower risk of both seminoma and nonseminoma testicular cancer. This is difficult to explain, but estimates were not statistically significant and based on only four publications and rather small populations (combined n_seminoma/nonseminoma_: 380/461) (Appendix A).

### 4.3. Quality and Risk of Bias of Included Publications

The included publications generally had a high completeness of reporting, but more than half of the publications were considered at a high risk of bias. These publications were particularly prone to information and residual confounding bias, with the latter being due to not accounting for confounders at all. As testicular cancer ascertainment was relatively comparable across the publications, the primary sources to information bias were identified through methods for exposure ascertainment. Most publications ascertained maternal cigarette smoking through self-reporting, with a large number of case-control studies relying on data collected at the time of the son’s testicular cancer diagnosis, thus after the occurrence of exposure, which is not necessarily a reliable measure due to the risk of recall bias. Given this, we expect the self-report of exposure will bias the estimate away from the null above 1. Similarly, cigarette smoking is recognized to be notoriously underreported, particularly by pregnant women, likely due to fear of stigmatization [39]. Importantly, only one publication included direct measurements of cigarette smoking exposure, hereunder cotinine, which is considered the gold standard as it is associated with minor risks of exposure misclassification.

The ecological studies were all large (n_range_: 4586–12,592), implying statistical precision. However, they utilized all lung cancer as a proxy for maternal exposure to cigarette smoking, and although tobacco smoking is indeed a predominant risk factor of lung cancer with about 90% of lung cancer cases being attributed to tobacco use [40], not all lung cancer cases are attributed to direct tobacco exposure, such as adenocarcinomas, which is the most common sub-type of non-smokers [41]. Thus, this method of exposure ascertainment could have resulted in some exposure ascertainment bias. In one instance, lung cancer was ascertained from the assumed maternal generations of the testicular cancer cases, thus inherent risk of unavoidable uncertainty was potentially attached to these generated risk estimates. It is also recognized that ecological studies are prone to ecological fallacy, and conclusions will only reflect the group and not the individual.

In contrast, the case-control study design was considered methodologically adequate for studying rare diseases and outcomes [42]; however, these were often hampered by relatively small sample sizes. Therefore, even in instances where an increased risk was indicated, the estimates did not reach statistical significance.

### 4.4. Exposure to Smoking in Pregnancy and Male Reproductive Health

Although our systematic review did not provide substantial evidence into an association between maternal exposure to cigarette smoking during pregnancy and testicular cancer, it remains uncertain whether our overall results are due to a lack of an association or perhaps due to the methodological limitations consistently observed across the included publications. Intriguingly, previous studies have found a link between maternal smoking during pregnancy and a higher risk of other parameters of male reproductive health, hereunder cryptorchidism [43,44], reduced sperm quality [45,46,47] and hypospadias [48,49], but also with contradictory results [44,50,51]. Nonetheless, cigarette smoke is known to contain several carcinogenic compounds [52] and components hereof, such as nicotine, have the ability to cross the placental barrier to the fetus [53]. Considering this broader evidence on male reproductive health and the massive positive trends seen in our qualitative trend synthesis, we cannot dismiss that a true effect may still be plausible.

### 4.5. Strengths, Limitations, and Added Value of Our Review

The main strength of our systematic review is the extensive and transparent literature search followed by the application of predefined eligibility and quality assessment tools. On balance, the present review provides an updated and deepened understanding about the available evidence on smoking during pregnancy and testicular cancer. Considering the prevalent behavior of active smoking during pregnancy [54] and the extent to which pregnant women are exposed to secondhand smoke [5], the link between maternal cigarette smoking and later testicular cancer in offspring remains critical to determine. An earlier systematic review [20] and a separate meta-analysis [37] previously evaluated the association but lacked a comprehensive discussion of the methodological limitations and a critical assessment of the risk of bias. We attempted to address this through our systematic examination of the quality and appropriateness of the methodology utilized for the included publications, thus forming proposals for future research. We further provided an in-depth review of testicular cancer as an outcome, in contrast with the latest systematic review that assessed a broader range of childhood reproductive health.

A limitation of our review includes the confined number of included publications. Our a priori aim was to provide an updated review given that the latest review was published almost a decade ago. To our surprise, no original research has been conducted since 2009, but in our opinion, this only emphasizes the need for further research using more optimal data and illuminating the methodological limitations previously highlighted. The lack of an association in the current systematic review calls for further research based on valid exposure data to provide substantial inference. Ideally, we sought to solely include publications utilizing the gold standard of cotinine measurements, but we were compelled to widen our review by including various proxies of exposure. Prospective cohort studies utilizing biological sampling are of a stronger design; however, such studies are difficult to implement. Finally, the evaluation of bias may potentially be prone to subjectivity, but this was to the best of our abilities avoided by involving two researchers who independently evaluated the publications, with any discrepancies resolved by a third researcher.

## 5. Conclusions

In this systematic review and meta-analysis, we did not find evidence of an association between maternal exposure to cigarette smoke and risk of testicular cancer in offspring. The qualitative trend synthesis suggested an overall positive trend but was hampered by statistical power and not confirmed in the meta-analysis. It is evident that there is a gap in studies assessing prenatal exposure to cigarette smoke utilizing more direct and precise methods. Given the mixed quality of the included publications and their methodological limitations, further original research based on valid exposure data is needed to fully establish the association.

## Figures and Tables

**Figure 1 life-13-00618-f001:**
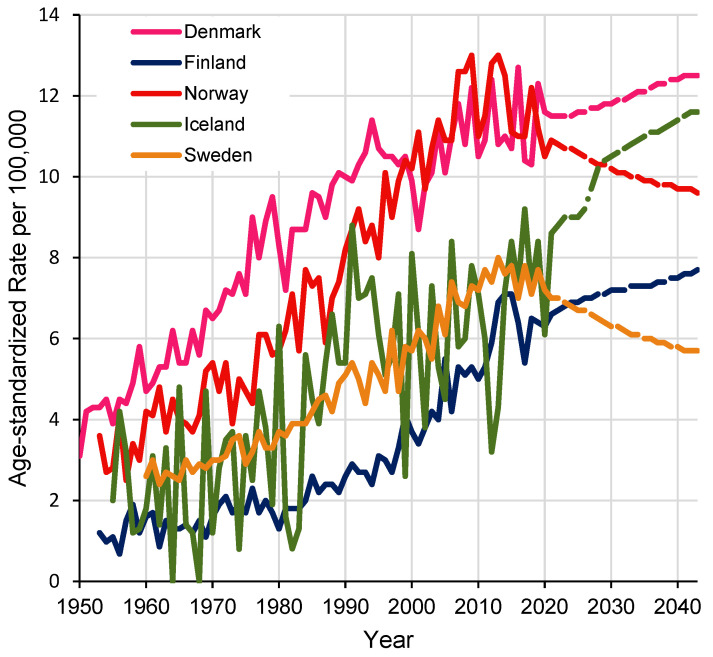
Age-standardized testicular cancer rates in Nordic countries (1950–2040) [12,13]. Created with NORDPRED software: http://www.kreftregisteret.no/en/Research/Projects/Nordpred/ accessed on 15 February 2023.

**Figure 2 life-13-00618-f002:**
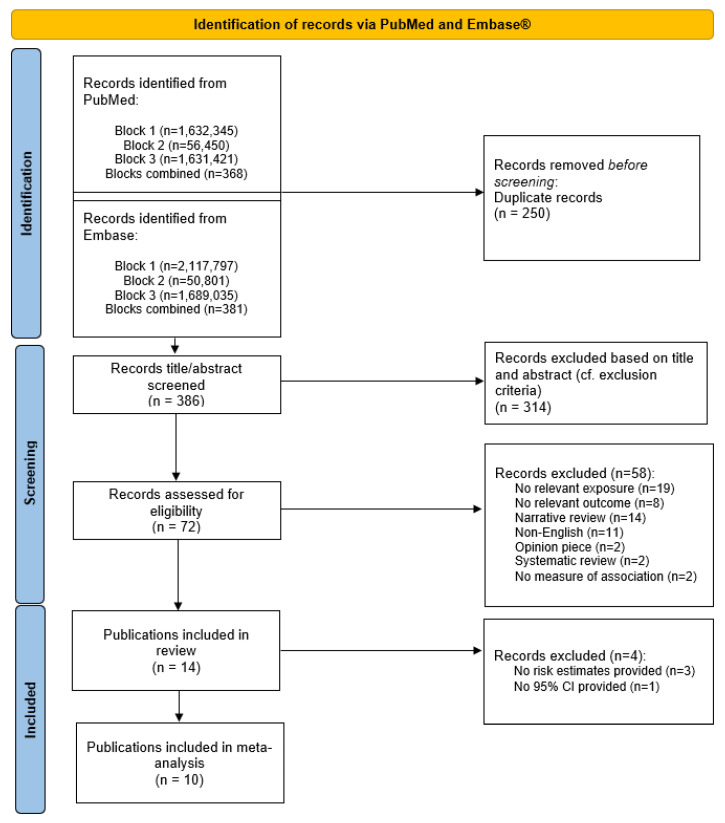
Flow diagram of the identified English articles published before 14 September 2022.

**Figure 3 life-13-00618-f003:**
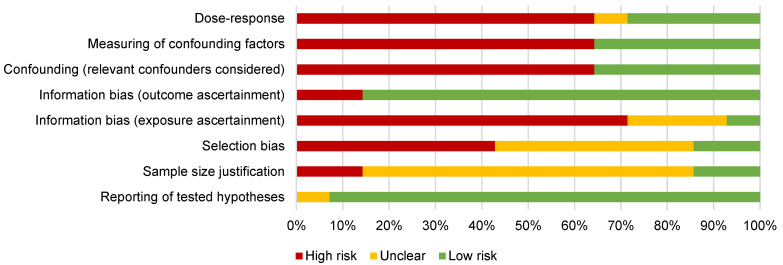
Risk of bias and confounding across the included publications (*n* = 14).

**Figure 4 life-13-00618-f004:**
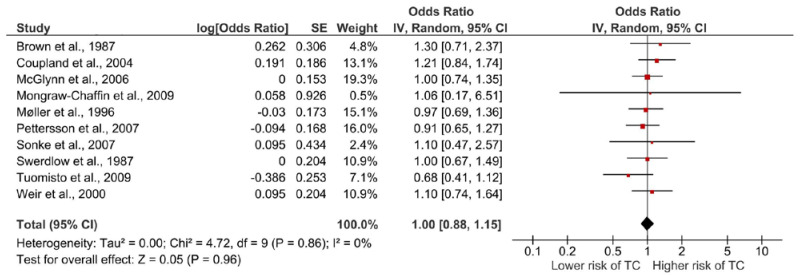
Forest plot of included studies providing a summary estimate of the association between assessing maternal exposure to cigarette smoke during pregnancy and risk of testicular cancer in offspring [25,26,30,31,32,33,35,36,37,38].

**Figure 5 life-13-00618-f005:**
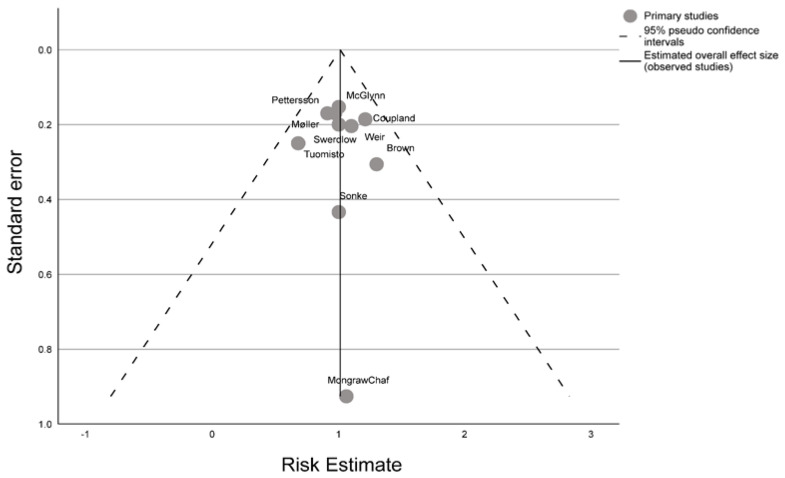
Funnel plot of risk estimates from included studies in the meta-analysis (*n* = 10) assessing maternal exposure to cigarette smoke and risk of testicular cancer in offspring.

**Table 1 life-13-00618-t001:** Study characteristics and qualitative trend synthesis for associations between maternal exposure to cigarette smoke and testicular cancer in offspring.

Author, Year	Country	Study Design	Study Period	Study Population (N)	Exposure Ascertainment	Outcome Ascertainment	Estimate ↓ Negative ↑ Positive ↔ None	CR	Bias	Meta-Analysis
Case-control studies										
Direct quantification of maternal exposure to cigarette smoke										
Tuomisto et al., 2009 [37]	Finland, Sweden, Iceland	Case-control	1985–2003 (Finland), 1976–2006 (Sweden), 1979–2006 (Iceland)	70/519: cases/controls	Cotinine measured in serum	National Cancer Registries ^a^	↓	7	6	Yes
Indirect quantification of maternal exposure to cigarette smoke										
Brown et al., 1986 [25]	USA	Case-control	1979–1981	271/259: cases/controls	Questionnaire by interview	Membership/registration at medical centers	↑	7	10	Yes
Coupland et al., 2004 [26]	UK	Case-control	1984–1987	447/522: cases/controls	Postal questionnaire	Cancer treatment centres and regional cancer registries, further confirmed by general practitioner notes	↑	8	6	Yes
Henderson et al., 1987 [28]	USA	Case-control	1972–1974	131/131: cases/controls	Questionnaire (retrospective)	Cancer surveillance programme	↔	8	12	No
McGlynn et al., 2006 [30]	USA	Case-control	2002–2005	754/928: cases/controls	Computer-assisted telephone interview	U.S. Servicemen’s Testicular Tumor Environmental and Endocrine Determinants study and Defense Medical Surveillance System ^a^	↔	8	6	Yes
Mongraw-Chaffin et al., 2009 [31]	USA	Case-control	1959–2003	20/60: cases/controls	Questionnaire	California Cancer Registry	↑	5	10	Yes
Møller et al., 1996 [32]	Denmark	Case-control	1989–1990	296/287: cases/controls	Questionnaire	Danish Cancer Registry	↓	8	9	Yes
Pettersson et al., 2007 [34]	Sweden	Case-control	1973–2002	192/494: cases/controls	The Swedish Medical Birth Register	Swedish National Cancer Registry ^a^	↓	8	4	Yes
Sonke et al., 2007 [35]	USA	Case-control	1990–1996	144/86: cases/controls	Questionnaire (cases/controls), interviews (mothers)	Registration at cancer center	↔	7	7	Yes
Swerdlow et al., 1987 [36]	UK	Case- control	1979–1981	218/404: cases/controls	Survey interview	Cancer registries, clinical department records, clinical staff, hospital diagnostic indexes, hospital activity analysis, and death certificates	↔	7	11	Yes
Weir et al., 2000 [38]	Canada	Case-control	1987–1989	325/490: cases/controls	Self-administered questionnaire	Ontario Cancer Registry ^a^	↑	7	5	Yes
Ecological studies										
Hemminki et al., 2005 [27]	Sweden	Ecological cohort	1958–2002	4586 cases	Cancer registry data	Cancer Registry data ^a^	↑	7	8	No
Kaijser et al., 2003 [29]	Sweden	Ecological cohort	1958–1997	12,592 sons (40 cases)	The Swedish Cancer Registry	Swedish Cancer Registry	↑ *	7	8	No
Pettersson et al., 2004 [34]	Nordic countries	Ecological correlation	1910–1968	Sweden (*n* = 55,930), Norway (not defined), Denmark (*n* = 34,018), and Finland (*n* = 2152).	Survey	Cancer Registries	↑ *	5	11	No

* Statistical significance (*p* < 0.05). ^a^ Assessed for histological subtype.

## Data Availability

All data are available for viewing in the article and Appendix A.

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
