# Peer review of "Maternal Exposure to Cigarette Smoke during Pregnancy and Testicular Cancer in Offspring: A Systematic Review and Meta-Analysis"

_life, 2023, doi:10.3390/life13030618_

Round 1

Reviewer 1 Report

The systematic review and meta-analysis by Astrid L. Beck & al. on maternal exposure to cigarette smoke during pregnancy and testicular cancer in offspring relates to the mystery of great variation in incidence of testicular cancer between countries and population subgroups within countries, and temporal trends in this variation. This has been discussed among epidemiologists for more than 60 years and still no explanation has been found.

One of the theories discussed over all decades has been maternal smoking. This systematic review nicely collects together studies based on this hypothesis and ends up to an overall summary risk estimate of 1.00 (95% CI: 0.88; 1.15) for maternal exposure to cigarette smoke and risk of testicular cancer in offspring. The authors conclude that their finding encourages further research based on valid exposure data – the same conclusion that has been repeated hundreds of times before.

The systematic review and the meta-estimates have been done in a proper way. This reviewer was not able to find the Figures and Tables which were supposed to be part of the article. From the Supplementary Figures it can however be seen that the presentation style is correct.

The writing style is a bit boring and would gain from interest-increasing illustrations. For instance, the Introduction shortly describes that there has been geographical differences and changes in in incidence over time, but the exceptional strength of such patterns is not really demonstrated. The Danish team could take an example of the world-record incidence in Denmark some decades ago and describe how the highest incidence rapidly moved to mid-Norway where the rate now is more than 5-fold as compared to Eastern Finland with rather similar way of life. There are, e.g., impressive map presentations on this issue which could be copied to add colour the Introduction (see Patama & al (2018) Small-area based map animations of cancer incidence in the Nordic countries, 1971-2015. Nordic Cancer Union; astra.cancer.fi/cancermaps/Nordic_18).

The authors use word “surprising” to characterize some of the observations. For a open-minded scientist nothing should be surprising and it is better to use word “unexpected”. One of the “surprises” if the finding that summary estimates for both seminoma and non-seminoma are <1 although the estimate for both types combined is 1.00. As the authors explain in their Discussion, this is well possible because the subtype-specific estimates are based on only four rather small studies which in addition come from population with atypical testicular cancer incidence. Therefore the subtype-specific estimates should not be highlighted in the Abstract.

The original aim of the systematic review was to update the conclusion of a meta-analysis from 2013, but it appeared that no novel results had been published after 2009 and hence the information basis of this meta-analysis is same as ten yours earlier. Still the new meta-analysis is worth publishing because it is a reminder of the need of not giving up in search of the testis cancer mystery.  

Author Response

Rebuttal: Point-by-point responses to reviewers 1 & 2

We thank Reviewer 1 and 2 for their comprehensive reviews and useful comments. We have responded to comments point-by-point below and incorporated all suggestions to the best of our abilities. Page and line numbers that we refer to in our responses correspond to the revised manuscript in track changes.

Reviewer 1

Comment 1: The systematic review and meta-analysis by Astrid L. Beck & al. on maternal exposure to cigarette smoke during pregnancy and testicular cancer in offspring relates to the mystery of great variation in incidence of testicular cancer between countries and population subgroups within countries, and temporal trends in this variation. This has been discussed among epidemiologists for more than 60 years and still no explanation has been found. One of the theories discussed over all decades has been maternal smoking. This systematic review nicely collects together studies based on this hypothesis and ends up to an overall summary risk estimate of 1.00 (95% CI: 0.88; 1.15) for maternal exposure to cigarette smoke and risk of testicular cancer in offspring. The authors conclude that their finding encourages further research based on valid exposure data – the same conclusion that has been repeated hundreds of times before.

The systematic review and the meta-estimates have been done in a proper way. This reviewer was not able to find the Figures and Tables which were supposed to be part of the article. From the Supplementary Figures it can however be seen that the presentation style is correct.

Response 1: Thank you for the appraisal of our paper. In total we submitted four main figures, one main table, two supplementary tables and two supplementary figures as separate files. These have now been added to the main document for your perusal. We apologize for the inconvenience and the confusion this may have caused.

Comment 2: The writing style is a bit boring and would gain from interest-increasing illustrations. For instance, the Introduction shortly describes that there has been geographical differences and changes in in incidence over time, but the exceptional strength of such patterns is not really demonstrated. The Danish team could take an example of the world-record incidence in Denmark some decades ago and describe how the highest incidence rapidly moved to mid-Norway where the rate now is more than 5-fold as compared to Eastern Finland with rather similar way of life. There are, e.g., impressive map presentations on this issue which could be copied to add colour the Introduction (see Patama & al (2018) Small-area based map animations of cancer incidence in the Nordic countries, 1971-2015. Nordic Cancer Union; astra.cancer.fi/cancermaps/Nordic_18).

Response 2: We thank Reviewer 1 for the useful source and have thus incorporated a descriptive color figure (page 4, line 25-26) depicting the age-standardized rates of testicular cancer in the Nordic countries to accommodate for the lack of captivating language and appearance of the manuscript.

Comment 3. The authors use word “surprising” to characterize some of the observations. For a open-minded scientist nothing should be surprising and it is better to use word “unexpected”. One of the “surprises” if the finding that summary estimates for both seminoma and non-seminoma are <1 although the estimate for both types combined is 1.00. As the authors explain in their Discussion, this is well possible because the subtype-specific estimates are based on only four rather small studies which in addition come from population with atypical testicular cancer incidence. Therefore the subtype-specific estimates should not be highlighted in the Abstract.

Response 3. We have changed the use of the word “surprisingly” to “unexpectedly” (page 18, line 307). We have retained the sub-specific estimates in the abstract as these lend a lot of novelty to the paper, and we have already specified in the abstract that these sub-specific estimates are based on fewer papers.

Comment 4. The original aim of the systematic review was to update the conclusion of a meta-analysis from 2013, but it appeared that no novel results had been published after 2009 and hence the information basis of this meta-analysis is same as ten yours earlier. Still the new meta-analysis is worth publishing because it is a reminder of the need of not giving up in search of the testis cancer mystery.

Response 4.  We thank the reviewer for this important point, and as we have described in our paper (page 4 [line 29-31] and page 20 [line 4358-360]) the previous systematic reviews from 2009 and 2014 lacked a comprehensive discussion of methodological limitations of the included studies, performed no trend analysis, did not conduct a critical assessment of risk of bias, and only one of the two moreover conducted a meta-analysis. In addition, the included papers in the aforementioned systematic reviews differ from ours. These elements are all essential when summarizing the validity of reported results. A very central message in our review was that there is a gap of studies assessing prenatal exposure to cigarette smoke utilizing more direct and precise methods, such as cotinine measurements in blood. Given the mixed quality of the included publications and their methodological limitations, further original research based on valid exposure data is needed to fully establish the association.

Reviewer 2 Report

1.In page 1, line 5-7

Is it necessary to write orcid number?

2. In page 2, line 51-52

This sentence should have proper reference.

3. page 2, line 56

Multiple references here should be reformed

4. Page 2, line 57 and 58

Middle- sentence reference and multiple references should be reconsidered in this part.

5.In page 2, line 60-61

The authors have written about testicular cancer here but this sentence has disrupted the consistency of the text because it has no logical link with its prior paragtaph. Please reform this section according to mentioned note.

6.In page 2, line 61

Why the authors have entered multiple references here?

7. In page 2, line 62-64

This part needs suitable reference.

8. Page 2, line 66, 67 and 69

Multiple references shiuld be reformed.

9. In page 2, line 74

Multiple reference and middle-sentence should be reformed.

10. In page 2, line 75-78

Why the authors have mentioned this part?

It seems that this part belongs to the part "Discussion"

11. In page 2, line 81-83

Is it necessary to mention this section here? If yes, why?

12. In page 2, line 86

Please reform middle-sentence reference

13. Why the manuscript have no figure or table in its main text?

14. In page 3, line 110 and 120

Is it not better to show "Eligibility criteria" and " Exclusion criteria " in a simple figure (or chart)?

15. In page 3, line 126, part "Selection of literature "

This part should be simplified and be demonstrated in the form of one simple chart or fugure

16. In page 3, line 136

Where is Fugure 1?

17. Page 3, line 140

Multiple reference should be reformed.

18. Why you have some supplementary tables but you do not have any tables in the main text of the manuscript?

It seems to be necessary to insert some clear scientific tabe(s) in order to make the manuscript more comprehendable.

19. Page 4, line 191

Where is Table 1?

20. Page 4, line 191

Why the authors have inserted reference here? And why multiple references?

21. In page 4, line 191-193

Why the authors have written this sentence here? Is it not belong to the part" material and methods" ?

22. In page 4 and 5, line 188-216

Why the authors have not simplify this part of the manuscript in the form of a simple figure or chart or table?

23. In page 5, line 216

Where is Figure 2?

24. Page 5, line 218

Where is Table 1?

25. Page 5, line 226-229

Why the authors have remarked this part in the section "Results"? It belongs to the part "Discussion" but if it is one of your findings, why you have inserted the reference?

26. In page 6 and 7, line 264-296

Why the authors have not compare their findings with other scientific studies in this section of the manuscript?

27. In page 7, line 318 and 325

Please reform middle-sentence reference

28. In page 7, line 335-337

Please reform all multiple and midde- sentence references in this part

29. Page 8, line 343-356

Reform all middle-sentence references

30. Please check reference list. (Especially the title)

Author Response

Rebuttal: Point-by-point responses to reviewers 1 & 2

We thank Reviewer 1 and 2 for their comprehensive reviews and useful comments. We have responded to comments point-by-point below and incorporated all suggestions to the best of our abilities. Page and line numbers that we refer to in our responses correspond to the revised manuscript in track changes.

Reviewer 2

General response to comments: Thank you for your comments and the various observations you have provided throughout the manuscript. It is greatly appreciated. We noted that a vast majority of your comments took basis in the reform of references. In some instances, your request for reform was uncertain, however, we have tried to the best of our abilities to fully understand what was meant by “reform” by assessing content, date, format, and placement of references in question. We hope our action is to your agreement, and if not, we are open to further, specific comments. We have responded to your comments point by point below.

Comment 1: In page 1, line 5-7. Is it necessary to write orcid number?
Response 1: The ORCiD number is a unique number that distinguishes individual researchers from one another, particularly of importance when researchers have a similar, or in some instances, the same name. The journal editorial office has included the ORCID numbers in our template version, so we assume these will be retained to uniquely identify us.

Comment 2: In page 2, line 51-52. This sentence should have proper reference.
Response 2: We have now incorporated a relevant reference (line 3, page 3).

Comment 3: Page 2, line 56. Multiple references here should be reformed
Response 4: We assume the request of reform takes basis in incorrect reference format. We have now accommodated the comment by adjusting the format (page 3, line 6).

Comment 4: Page 2, line 57 and 58. Middle- sentence reference and multiple references should be reconsidered in this part.
Response 4: The two references included here are both pertinent and new and have been retained: Reitsma et al. Spatial, temporal, and demographic patterns in prevalence of smoking tobacco use and initiation among young people in 204 countries and territories, 1990- 2019 (2021) and Lange et al. National, regional, and global prevalence of smoking during pregnancy in the general population: a systematic review and meta-analysis (2018). The statements made in line 57 and 58 on page 2 moreover derive from the following references and thus it is assumed that Comment 4 does not concern reference content and subsequently this requires no further action.

Comment 5: In page 2, line 60-61. The authors have written about testicular cancer here but this sentence has disrupted the consistency of the text because it has no logical link with its prior paragtaph. Please reform this section according to mentioned note.
Response 5: Thank you for this observation. We agree there is no coherency between the two paragraphs and have thus adjusted accordingly by adding “As pioneered by Clemmesen almost three decades ago, prenatal exposure to cigarette smoke is a potential health risk proposed to be related to testicular cancer in offspring” (page 3, line 10-12).

Comment 6: In page 2, line 61. Why the authors have entered multiple references here?
Response 6: In page 2, line 61, there are two references: Gaddam et al. Testicle Cancer (2022) and Giona. The Epidemiology of Testicular Cancer (2022). Two references are added here as two age spans are predominantly reported in the literature, 15-40 and 15-44 years. We tried to accommodate for that variety by combining the two age spans into one, 15-44 years, and furthermore support our argument with the two references. Thus, these multiple and relevant references are retained here.

Comment 7: In page 2, line 62-64. This part needs suitable reference.
Response 7: References were previously lacking, so we have now incorporated a relevant reference (page 3, line 16).  

Comment 8: Page 2, line 66, 67 and 69. Multiple references should be reformed.
Response 8: We assume Reviewer 2’s request for reform takes basis in the misplacement of references. We have thus accommodated for this by correcting the placement of references (page 3, line 21). In regard to the two other references in question, it is uncertain to us what is required to reform as per Reviewer 2’s suggestion. 

Comment 9: In page 2, line 74. Multiple reference and middle-sentence should be reformed.
Response 9: Multiple references are included here to support the statement of “several epidemiological studies” at the start of the sentence. It is moreover uncertain what Reviewer 2 means by “reformed” and we have thus retained these important references (page 4, line 28).   

Comment 10: In page 2, line 75-78. Why the authors have mentioned this part? It seems that this part belongs to the part "Discussion"
Response 10: There is presently consensus regarding the uncertainty relating to the association between exposure to maternal smoking during pregnancy and testicular cancer in offspring, thus, we wanted to emphasize the added value of this current study in our introduction. Thus, due to the importance of this information in our introduction to our present review, we have retained this sentence in the introduction.

Comment 11: In page 2, line 81-83. Is it necessary to mention this section here? If yes, why?
Response 11: A preliminary literature search highlighted the need for studies utilizing valid exposure data and thus we added this novel and important assessment of the included papers’ risk of bias as an additional research aim.

Comment 12: In page 2, line 86. Please reform middle-sentence reference
Response 12: The systematic review was conducted according to PRISMA guidelines for systematic reviews, and therefore we have elected to keep it there (page 5, line 39).

Comment 13: Why the manuscript have no figure or table in its main text?
Response 13: According to the journal’s author instructions, we were allowed to attach the figures and tables as a separate document and therefore we assumed they would place them in the manuscript according to their preferred format – as all other journals do. In total we submitted four main figures, one main table, two supplementary tables and two supplementary figures as separate files. These have now been added to the main document for your perusal. We apologize for the inconvenience and the confusion this may have caused.

Comment 14: In page 3, line 110 and 120. Is it not better to show "Eligibility criteria" and " Exclusion criteria " in a simple figure (or chart)?
Response 14: We apologize that you were unable to see this figure. As stated in response 13, we have now placed this figure in the main text for your perusal (page 8, line 102-144).

Comment 15: In page 3, line 126, part "Selection of literature ". This part should be simplified and be demonstrated in the form of one simple chart or fugure
Response 15: Kindly see response 13. This has been accommodated for by placing the figure in the main text (page 8, line 102-144).

Comment 16: In page 3, line 136. Where is Fugure 1?
Response 16: Kindly see previous response 13.

Comment 17: Page 3, line 140. Multiple reference should be reformed.
Response 17: It is uncertain what the reviewer is referring to. The references are formatted according to the journal requirements.

Comment 18: Why you have some supplementary tables but you do not have any tables in the main text of the manuscript? It seems to be necessary to insert some clear scientific tabe(s) in order to make the manuscript more comprehendable.
Response 18: Kindly see previous response 13.

Comment 19: Page 4, line 191. Where is Table 1?
Response 20: Kindly see previous response 13.

Comment 20: Page 4, line 191. Why the authors have inserted reference here? And why multiple references?
Response 20: Here specifically we are referring to all included papers in the systematic review. Not all included papers are included in the meta-analysis, so as a helping hand to the reader throughout the main text, we differentiate between these two groups of papers by referencing them, so the reader knows which exact papers we are referring to. 

Comment 21: In page 4, line 191-193. Why the authors have written this sentence here? Is it not belong to the part" material and methods" ?
Response 21: The date in which the papers are published as well as their study populations are characteristics of theirs, which we believe should be described and emphasized as results of the review – and what characterizes the evidence within the field. We want to give the reader an idea of how potentially outdated findings may be. Moreover, the papers’ dates are collected from the data extraction process which happens post establishment of methods and the findings generated hereof are results, therefore we consider it befitting to include in the results section and not methods.

Comment 22. In page 4 and 5, line 188-216. Why the authors have not simplify this part of the manuscript in the form of a simple figure or chart or table?
Response 22: Kindly see previous response 13.

Comment 23: In page 5, line 216. Where is Figure 2?
Response 23: Kindly see previous response 13.

Comment 24: Page 5, line 218. Where is Table 1?
Response 24: Kindly see previous response 13.

Comment 25: Page 5, line 226-229. Why the authors have remarked this part in the section "Results"? It belongs to the part "Discussion" but if it is one of your findings, why you have inserted the reference?
Response 25: We agree with Reviewer 2 that there should be no reference here and have thus removed the reference (page 15, line 235).

Comment 26: In page 6 and 7, line 264-296. Why the authors have not compare their findings with other scientific studies in this section of the manuscript?
Response 26: We compare our findings further down in discussion, more specifically page 20 (line 358-360). We were only able to compare with two similar studies.

Comment 27: In page 7, line 318 and 325. Please reform middle-sentence reference
Response 27: It is uncertain what the reviewer is referring to. The references are formatted according to the journal requirements.

Comment 28: In page 7, line 335-337. Please reform all multiple and midde- sentence references in this part.
Response 28: : It is uncertain what the reviewer is referring to. The references are formatted according to the journal requirements. We assume that the reviewer is suggesting that all references included in these lines should be placed at the end of our statement. This is not an incorrect way. However, nor is the way we have done it. We have thus retained them.

Comment 29: Page 8, line 343-356. Reform all middle-sentence references
Response 29: : It is uncertain what the reviewer is referring to. The references are formatted according to the journal requirements. As previously mentioned in response 28, the placement of references can be subjective, and as the references follow after a statement in these lines here, we have chosen to retain them.

Comment 30: Please check reference list. (Especially the title)
Response 30: The journal’s author restrictions refer to the bibliography/references as “References”, thus we have chosen the tile of this section according to their guidelines. We have moreover used a bibliography software package (Endnote, to be precise) and the journal’s preferred reference style.

Round 2

Reviewer 2 Report

After the corrections, the article is suitable for publication in the journal. I have no additional suggestions.